# Socio-Economic Analysis of the Construction and Building Materials' Usage—Ecological Awareness in the Case of Serbia

**Milica Vidak Vasić** [1],*, **Gaurav Goel** [2], **Mandefrot Dubale** [3], **Slavica Živković** [4], **Milan Trivunić** [5], **Milada Pezo** [6] and **Lato Pezo** [7]

1   Laboratory for Building Ceramics, Institute for Testing of Materials IMS, Bulevar Vojvode Mišića 43, 11000 Belgrade, Serbia
2   School of Energy and Environment, Thapar Institute of Engineering Technology, Patiala 147004, India
3   Department of Civil Engineering, Indian Institute of Technology Guwahati (IITG), Guwahati 781039, India
4   Institute for Testing of Materials IMS, Bulevar Vojvode Mišića 43, 11000 Belgrade, Serbia
5   Department of Civil Engineering and Geodesy, Faculty of Technical Sciences, University of Novi Sad, Trg Dositeja Obradovića 6, 21000 Novi Sad, Serbia
6   Department of Thermal Engineering and Energy, "VINČA" Institute of Nuclear Sciences, University of Belgrade, 11001 Belgrade, Serbia
7   Institute of General and Physical Chemistry, University of Belgrade, Studentski trg 12, 11000 Belgrade, Serbia
*   Correspondence: milica.vasic@institutims.rs

**Abstract:** The main idea of the present study was to investigate the impact of the ongoing world crisis on the socio-economic issues in Serbia concerning building materials usage and purchase. This research fills in the gaps in the literature and contributes to the comprehension of how the crisis has affected salaries, market pricing, and materials consumption in the building sector. The data are gathered through a questionnaire and analyzed using a statistical methodology (frequencies, descriptive statistics, and Spearman's correlations). Most of the former studies investigated the surveys conducted on specialists in the field, while this study analyzed the perspectives of random people. Socio-demographic issues are analyzed along with materials consumption before and after the crisis. A special emphasis is given to ecological awareness and novel materials usage. Additionally, it captures a broad shift in the economy and ecological consciousness in a developing country. The majority of respondents are open to using novel building materials and products, but their choice would largely be influenced by cost, the amount of effort involved, and their understanding of the advantages. Statistical approaches revealed that the crisis has a considerable impact on the markets for construction and building supplies, altering consumers' decisions when purchasing. This contribution lays the groundwork for developing countries in the modern world to improve sustainability and adopt circular thinking. Professionals in Serbia need to have a more eco-aware mindset and enhance how they provide pertinent information to potential clients. This study is limited by the number of respondents. For future mathematical modeling and forecasting, more answerers are needed.

**Keywords:** construction and building materials; sustainability; green materials; socio-economic survey; statistical analysis; developing country; Serbia; world crisis

## 1. Introduction

Serbia was in a moderate economical position in 2019, with real GDP growth of 3.2% and the lowest unemployment level in the last decade of 10.5%. The country was in a rather good financial situation in the early months of 2020 because it had reduced its external debt and significantly reduced its fiscal deficit [1]. Beginning in 2020, an unforeseen global pandemic swept the globe. Millions of people have died as a result of the coronavirus (COVID-19), which has spread to all continents. The ongoing fundamental changes in the economy since that time [2] persist through the war between Russia and Ukraine. In Serbia,

everyone was affected by the COVID-19 crisis, but workers in smaller businesses and in the informal economy were those that were most negatively impacted. However, within this framework, the construction industry did not suffer in revenue [1].

### 1.1. Environmental Framework in Construction and Building Materials and Structures

Given the rapidly deteriorating environmental health of the planet and the growing problem of waste and global warming, it is necessary to change the consciousness from the bottom up. One of the many leading problems to actively address is within the construction industry due to the intensive material, energy, and flue gas flow that has had a disastrous effect on the environment [3–10]. It is necessary to intensify the use of recycled materials while taking care to obtain products with good insulating properties, longer durability, and resistance to aggressive environments. Furthermore, 2050 is the target year for Roadmaps set by the European Commission, which set ambitious long-term goals for resource efficiency, energy savings, and a low-carbon economy [11,12]. Developing countries usually lag in improving environmental issues [13], so it is necessary to examine the perspectives and perceptions of respondents to be able to work towards general progress.

The construction industry is an exhaustive consumer of raw materials and energy, emitting up to 1% of global carbon and 36% of the total greenhouse gasses [3,11,12]. While construction and demolition wastes are estimated to be 36% in the EU [4], they are recovered by only about 50% on average [5]. Most of this waste belongs to concrete and ceramics [6], which is considered an accelerating environmental threat [7]. One of the primary difficulties is proper data collection on quantity [8]. The awareness of the legislation on construction and demolition waste needs to be improved, especially for micro and small construction companies [9]. Furthermore, the reuse of construction and demolition waste in novel recycling plants needs to become economically beneficial for the uptake by the industry [10].

Increasing the sustainability of the construction industry is intensively studied with the aims of using various kinds of secondary raw materials [14–17], introducing novel processes such as geopolymerization [18,19], and reusing the construction and demolition waste after the service life of the products [20,21]. Among other demands, increased thermal and sound insulation is very important. Despite the tremendous efforts implemented in laboratory scientific studies, industrial-scale data are lacking. The reason behind this is that the industrial activities and development of new products are often not oriented toward open publishing. Another problem is the quantity of the specific waste available for continuous production, which is the major concern of producers in Serbia. One of the rare examples in industrial examinations is the use of waste coal dust in traditional ceramics [22]. Moreover, large-scale glass-reinforced pipe samples incorporated with fly ash are also tested [23]. Additionally, roof tiles are produced by replacing raw clay with waste rice husk ash and ceramic sludge [24]. In addition, the waste packaging glass [25] is used as a flux (partial substitution of natural feldspar) in ceramic mass in industrial-sized samples.

There are many fascinating examples of cutting-edge construction and building eco-products worldwide. Densified pet bottles to build a load-bearing construction product are produced in Switzerland, and fire-retardant newspaper wood meant for flooring is produced in the Netherlands [26]. In addition, an example of densified waste materials is tiles produced from disposed coffee ground waste in Spain [26]; among reconfigured waste materials, there are the roof coverings made of Tetra Pak cartoons produced in India [26]. In addition, various forms of plastic waste are used as partial replacements for sand and gravel in asphalt mixtures in the USA. Agricultural byproducts are raw materials are used to create load-bearing and insulating panels [26]. In addition, wooden structures are regaining more attention as an effective balancer of the carbon footprint [27]. A 100-m-tall wooden residential tower is planned to be built in the next four years near Zürich [28]. There are numerous examples of eco-building materials that are not known to most people. Therefore, an additional challenge for the building sector, besides innovation

in green materials and products, is to educate and inform the public and professionals [29]. Through this, a long-term competitive advantage for the producers will be granted. The only relevant evaluation related to environmental issues is the conformity to Environmental Product Declaration (EPD) that is obtained by Life Cycle Assessment [30].

*1.2. Survey Questionnaires Review and the Knowledge Gap*

Because the majority of studies are focused on construction projects in developing nations [31,32], there are not many surveys that have been published thus far in the field of construction and building materials and goods.

In a study involving 60 Turkish manufacturers of construction materials, the readiness for innovation was examined. The results showed that this level is low regardless of respondents' ages, but rises in globally focused businesses [33]. Eze et al. [29] analyzed the green building materials market in Nigeria while including only the professionals as the respondents. Despite the widespread knowledge of the existence of environmentally friendly building materials, adoption remains low. Certainly, the market's lack of plenty of these materials is to blame as well. Another questionnaire survey by Michalak and Michałowski [34] is performed with respondents from the professional network in Poland. The inquiries focused on environment-related problems in the industry; the primary takeaways are that there is a critical need for higher education, starting with those who work in this sector. Potential investors in eco-friendly home construction were polled in Slovakia, and the results showed that lack of information is the biggest barrier keeping them from using old technology embraced by cutting-edge techniques such as rammed earth or straw bale construction [35].

The objective of this study was to examine the effects of the world crisis on socio-economic concerns, using Serbia as an example of a developing nation concerning the use and purchase of building materials and products. Furthermore, the study closes a gap in the literature and contributes to understanding how the crisis has affected wages, market pricing, and materials used in the construction industry. Additionally, it offers a broad overview of the environmental awareness and consumer acceptance of newly developed sustainable products. This research includes a socio-economic analysis of the developing country's market concerning the age of the respondents, the level of completed education, and income satisfaction. The respondents were chosen at random to ensure the diversity of the sample and prevent biases. Mutual comparisons of the period before and after the beginning of the crisis with the COVID-19 pandemic and socio-economic impact are present. This is the first study on this topic in the construction sector and is expected to contribute towards policy initiatives. The analysis of the obtained results is performed by using a statistical methodology (frequencies, descriptive statistics, and Spearman's correlations). The study is limited by the number of respondents. For future mathematical modeling and forecasting, more answerers are needed.

## 2. Data Collection and Methodology

*2.1. Data Collection*

To determine the state of the problems addressed in construction and building material choices and ecological awareness in Serbia, an extensive assessment of the pertinent literature was first carried out (Figure 1). Then, a questionnaire was created as the primary research tool to fill the knowledge gap. A survey containing 28 questions was designed, which was discussed by all authors of this paper living in developing countries. A draft of the questionnaire was presented to industrial and business experts. A small random sample of participants completed the survey after the agreed-upon revisions had been integrated to assure clarity, correct the final version, and increase the study's validity [36,37]. The list of questions is presented in Appendix A.

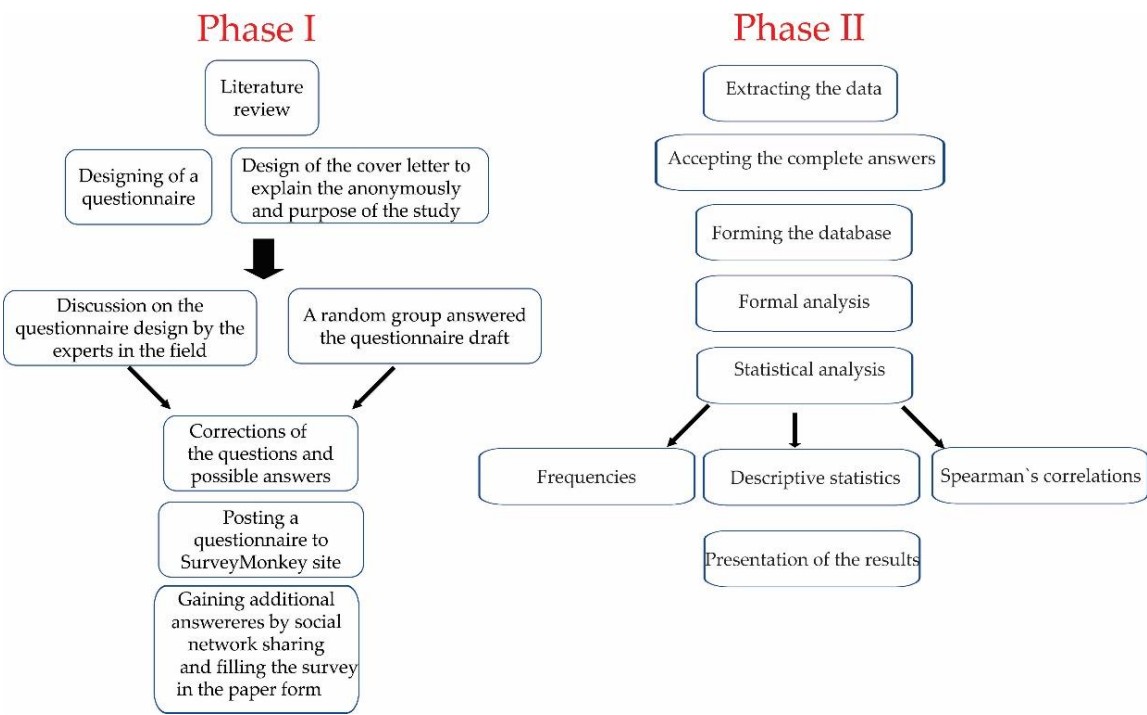

**Figure 1.** Flowchart of the study.

The minimal sample size to describe the population of 7 million in Serbia is calculated using the formula [38]:

$$n = \frac{Z^2 * p * (1-p)}{d^2} \tag{1}$$

where $n$ is the sample size, Z is the two-sided Z-score giving the 95% of confidence level (Z = 1.96), $p$ is the maximum probability of variation in the distribution ($p$ = 0.05), and $d$ is the margin of error–desired level of precision ($d$ = ±0.05). The minimal sample size is 385.

The calculation of the sample size for the finite surveys number ($n'$) was performed using the following formula:

$$n' = \frac{n}{1 + \frac{n}{N}}. \tag{2}$$

Furthermore, the confidence interval for a finite population parameter was calculated according to following equation:

$$CI' = p \pm z \cdot \sqrt{\frac{p \cdot (1-p)}{n'} \cdot \frac{N - n'}{N - 1}} \tag{3}$$

Based on this calculation, the confidence interval was between 0.450 and 0.550.

All residents aged between 18 and 80 living in Serbia were eligible to answer the questionnaire. The idea was to compile a random sample of people representing diverse age demographics, occupations, areas of expertise, and income satisfaction levels. It was also meant to elicit comments from both groups of respondents, including those who rent housing and those who, to a significant degree, own apartments, houses, or cottages. All participants from the same household were allowed to participate in the survey, as they reflected personal perceptions, impressions and experiences on relevant issues. Completing the survey was voluntary, but because the answers were gathered on a friendly basis, people who were not involved in construction activities responded, which provided a more accurate and comprehensive picture of the situation in Serbia. The survey was preceded by a cover letter detailing the objectives and specifics of this anonymous research. The survey was posted on the SurveyMonkey website, and the link was shared on social media

and with the author's personal and professional contacts to garner as much interest and variety as possible. Responses to the survey were accepted between January and April 2022. Additional respondents were individually gathered from June to October 2022 to complete the questionnaire in paper form.

The first set of 14 questions focused on vital sociodemographic data such as age, gender, education, profession (job description), and income satisfaction. The respondents were also questioned about the kind of home they occupied and if they owned a residential or a guest house. The purchasing of building materials before and during pandemics and global crises was the subject of the second set of 9 questions. Where appropriate, a quantitative seven-point Likert scale questionnaire was administered during this block. The final series, containing 5 questions, sought to ascertain whether respondents had used novel building materials and how likely they were to do so in the future.

The primary goal of this study was to pinpoint the critical elements that influence the use of cutting-edge ecological building products and materials. However, the study's scope was broader, and it also examined the variations in wages and prices in the years before and after the crisis that began with the pandemic, as well as the extent to which respondents carried out domestic building work during both times. The products concerned were split among building materials (cement, glue, paint, etc.) and building products of a definite shape (bricks, tiles, sanitary ware, carpentry, etc.). All the people above 18 and not older than 80 were eligible to express their experiences and thoughts. The factors that could have an impact on the use of building materials were used to design the questions.

*2.2. Statistical Analysis*

The chosen method to explain and quantify the outcomes was a statistical analysis of the collected data. The IBM SPSS 22 program was used to attain a basic grasp of the trends of extended sociodemographic data influence on the purchase of construction and building supplies and products. Exploratory data analysis (frequencies and descriptive statistics) was employed to obtain the general overview and distributions of the collected data [39].

Spearman's rank correlation coefficients were used to analyze the obtained database. Spearman's rank correlation, a non-parametric test that is frequently used to assess the level of agreement between two sets of ranking, was used to find the correlations between the discovered groups [37]. The results of the Spearman's rank correlation coefficient ($\rho$) are $-1 \leq \rho \leq +1$, with negative values associated with an inverse association and 0 denoting no correlation. To show the agreement between the answers in the questionary survey, the Spearman's rank correlation coefficients were determined. Moreover, a statistical significance of the obtained coefficients is taken into account. The Spearman's correlation coefficient is calculated using the following equation [40]:

$$\rho = 1 - \frac{6 \Sigma D^2}{n(n^2 - 1)}. \tag{4}$$

where $D$ is the difference in the ranks of the two variables compared, and $n$ is number of answerers.

Only respondents that fully answered the survey were included in the analysis. A total of 391 respondents were deemed qualified to describe a country such as Serbia, which has a population of 7 million people, with a confidence level of 95%. A 7-point Likert scale is used to adequately rank the experiences and impressions of respondents concerning personal questions because it is found to be accurate, easy to use, and a better reflection of a respondent's evaluation compared to a 5-point scale [41].

## 3. Results and Discussion

*3.1. Sociodemographic Data of the Tested Group*

Most of the respondents (60.1%) belonged to the age group of 31–50 (Table 1). The highest proportion of the participants was females with a university degree living in Belgrade, Serbia. The smallest share was respondents who completed college and primary school

(1.0 and 2.8%, respectively). The group mainly consisted of engineers (16.9%), medical workers (14.3%), and office workers (11.5%), as well as professors/lecturers (10.7%) and scientists (8.4%). A respectable number of various professions and unemployed people were present, adding diversity to the database that had been compiled. In a questionnaire study conducted in Croatia, respondents had a comparable sociodemographic distribution [41].

**Table 1.** Basic sociodemographic data and economic variables frequencies.

| The Basic Information on the Respondents | | Answer Code | Frequency | % |
|---|---|---|---|---|
| **Method of responding (Q1)** [1] | Smartphone | 1 | 281 | 71.9 |
| | Desktop | 2 | 76 | 19.4 |
| | Tablet | 3 | 2 | 0.5 |
| | In paper | 4 | 32 | 8.2 |
| **Age group (Q2)** | 18–30 | 18 | 61 | 15.6 |
| | 31–40 | 31 | 118 | 30.2 |
| | 41–50 | 41 | 117 | 29.9 |
| | 51–60 | 51 | 67 | 17.1 |
| | 61–70 | 61 | 27 | 6.9 |
| | 71–80 | 71 | 1 | 0.3 |
| **Gender (Q3)** | Female | 10 | 242 | 61.9 |
| | Male | 20 | 149 | 38.1 |
| **The number of inhabitants (Q4)** | Under 100,000 | 90,000 | 36 | 9.2 |
| | 100,000–300,000 | 100,000 | 30 | 7.7 |
| | 300,000–500,000 | 300,000 | 34 | 8.7 |
| | 500,000–800,000 | 500,000 | 5 | 1.3 |
| | 800,000–1,000,000 | 800,000 | 10 | 2.6 |
| | 1,000,000–2,000,000 | 1,000,000 | 12 | 3.1 |
| | Over 2,000,000 | 2,100,000 | 264 | 67.4 |
| **Education (Q5)** | Primary school | 1 | 11 | 2.8 |
| | High school | 2 | 77 | 19.7 |
| | College | 3 | 4 | 1.0 |
| | University | 4 | 179 | 45.8 |
| | Researcher/ PhD | 5 | 96 | 24.6 |
| | Professor | 6 | 24 | 6.1 |
| **Job description (Q6)** | Unemployed | 11 | 19 | 4.9 |
| | Manual worker | 12 | 20 | 5.1 |
| | Office worker | 13 | 45 | 11.5 |
| | Laboratory technician | 14 | 33 | 8.4 |
| | Medical personnel | 15 | 56 | 14.3 |
| | Craftsman | 16 | 12 | 3.1 |
| | Student | 17 | 10 | 2.6 |
| | Artist | 18 | 2 | 0.5 |
| | Engineer | 19 | 66 | 16.9 |
| | Scientist | 20 | 33 | 8.4 |
| | Lecturer/Professor | 21 | 42 | 10.7 |
| | Retired | 22 | 11 | 2.8 |
| | Manager | 23 | 22 | 5.6 |
| | Others (high school degree) | 24 | 10 | 2.6 |
| | Others (university degree) | 25 | 10 | 2.6 |
| **Income satisfaction (Q7)** | Very low satisfaction | 1 | 54 | 13.8 |
| | Low satisfaction | 2 | 24 | 6.2 |
| | Lower than average | 3 | 41 | 10.5 |
| | Average satisfaction | 4 | 108 | 27.6 |
| | Greater than average | 5 | 97 | 24.8 |
| | High satisfaction | 6 | 44 | 11.2 |
| | Extremely high satisfaction | 7 | 23 | 5.9 |

**Table 1.** *Cont.*

| The Basic Information on the Respondents | Answer Code | Frequency | % |
|---|---|---|---|
| | Decreased | 15 | 60 | 15.3 |
| **Changes in income during the** | Increased | 16 | 201 | 51.4 |
| **pandemic and the world crisis (Q8)** | Haven't changed | 17 | 105 | 26.8 |
| | Not applicable | 18 | 25 | 6.4 |

[1] Q—question (Appendix A).

Salary satisfaction is measured using a 7-level Likert scale, and the respondents mainly reported average (27.6%) or greater than average (24.8%) levels. After the beginning of the pandemic and the new crisis, their income mostly increased (51.4%) or did not change (26.8%). These increases may be due to achieving a higher position in the companies, which was not taken into account in this study. The coding was used to define the answers and perform the statistical analysis of the responses (Table 1).

The main residential unit was a family-owned apartment in a building 40 years old or older (Table 2). Most participants never moved or have relocated relatively recently (in the last 5 to 10 years). The majority of respondents do not own a rest private house, and among those that do, the majority of the structures are older than 40 years.

**Table 2.** Sociodemographic data related to real estate.

| The Information on the Respondents | | Answer Code | Frequency | % |
|---|---|---|---|---|
| | In the last 5 years | 5 | 93 | 23.8 |
| | During the last 10 years | 10 | 80 | 20.5 |
| **Change in the place of residence** | 20 years ago | 20 | 60 | 15.3 |
| **(Q9)** [1] | 30 years ago | 30 | 39 | 10.0 |
| | 40 years ago | 40 | 18 | 4.6 |
| | Never | 100 | 101 | 25.8 |
| **Apartment or a house** | Apartment/Hostel | 1500 | 268 | 68.5 |
| **(Q10)** | House | 2000 | 123 | 31.5 |
| **Is it rented?** | Yes | 1700 | 93 | 23.8 |
| **(Q11)** | No | 2700 | 298 | 76.2 |
| | In the last 5 years | 5 | 73 | 18.7 |
| **When was the building or the house built?** | During the last 10 years | 10 | 18 | 4.6 |
| | 20 years ago | 20 | 71 | 18.2 |
| **(Q12)** | 30 years ago | 30 | 61 | 15.6 |
| | 40 years ago | 40 | 168 | 43.0 |
| **Owning a cottage/rest house** | No | 500 | 252 | 64.4 |
| **(Q13)** | Yes | 1000 | 139 | 35.6 |
| | Not applicable | 1 | 235 | 60.1 |
| | In the last 5 years | 5 | 21 | 5.4 |
| **When was the rest house built?** | During the last 10 years | 10 | 28 | 7.2 |
| **(Q14)** | 20 years ago | 20 | 28 | 7.2 |
| | 30 years ago | 30 | 30 | 7.7 |
| | 40 years ago | 40 | 49 | 12.5 |

[1] Q—question (Appendix A).

### 3.2. Purchasing of Building Materials and Products

The overall perception among the respondents was that the price of a product is averagely or slightly more dependent on its quality (Figure 2a), that the price/quality relationship has somewhat altered since the crisis began with the pandemic (Figure 2b), and that the cost in construction and building industries has increased dramatically (Figure 2c).

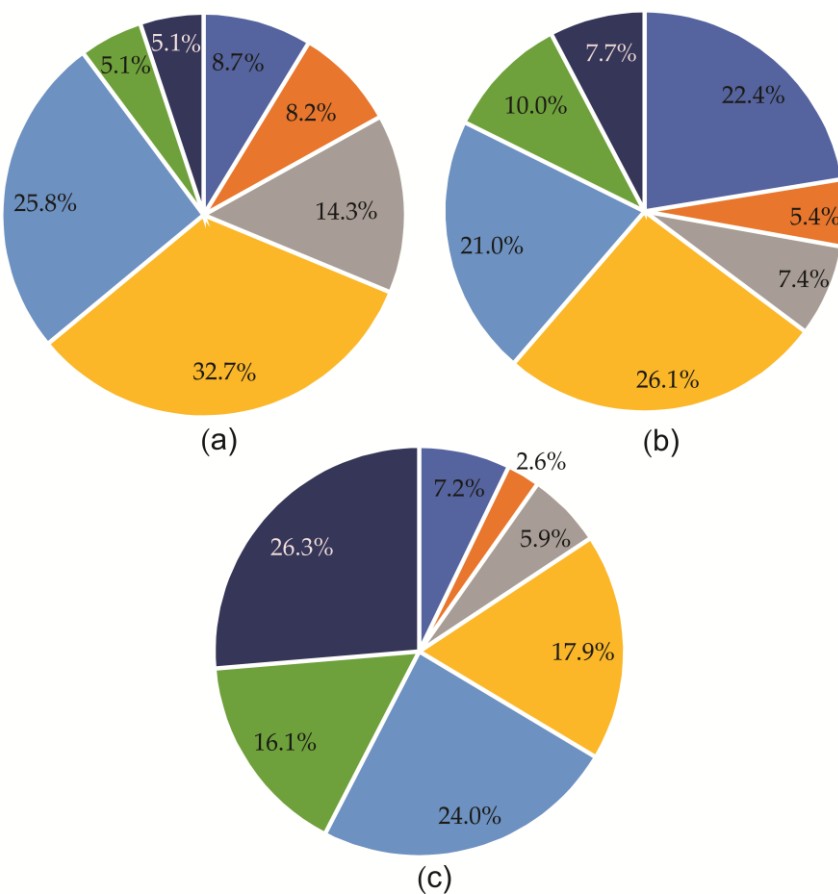

**Figure 2.** The 7-point Likert's scale (1—very low, 7—extremely high) that explains the percentage of perceptions on (**a**) the extent of price speaking of the products' quality (Q15), (**b**) price/quality ratio changes since the world crisis (Q22), (**c**) price changes since 2020 (Q23).

Data from manufacturers of building materials—which are the results of market research by official sales and thus the most relevant—show that costs in Serbia have dramatically risen since 2020. There are various factors, but among the earliest ones were border closures, the introduction of COVID passes, and delays in the delivery of vital raw materials [2], which significantly increased the cost of production. A further increase is anticipated by the rising cost of energy and the fact that it contributes to around 40% of the price of the final building product. In the following years, an increase of 40–60% in the price of electricity is expected. After this, the price of packaging dramatically increased by 20%, and the rise in online shopping has been cited as the cause. Transporting raw materials and final products now costs more due to rising fuel prices. While wages remain static, inflation started in 2020. All of this, along with the high level of interest in real estate investing since that time, led to a large increase in the price per square meter for residential space. Top-down and bottom-up economic shocks occurred and substantial changes in corporate and personal circumstances affected both domestic and global supply and demand trends for products and services [2].

Shortly before the pandemic, the majority of respondents (41.5%) had been purchasing building materials and products regularly, while many others (40.0%), primarily younger people living with their families, had not (Figure 3a). An increased percentage of study participants stopped buying the products of concern once the pandemic and crisis started (Figure 3b), which conflicts with the producers' market research in Serbia. The market research's selection of respondents had an impact on this discrepancy. The sales service analyzed potential clients, but in this study, all of the respondents were eager participants. Consequently, a more thorough picture of the situation in the analyzed country was created.

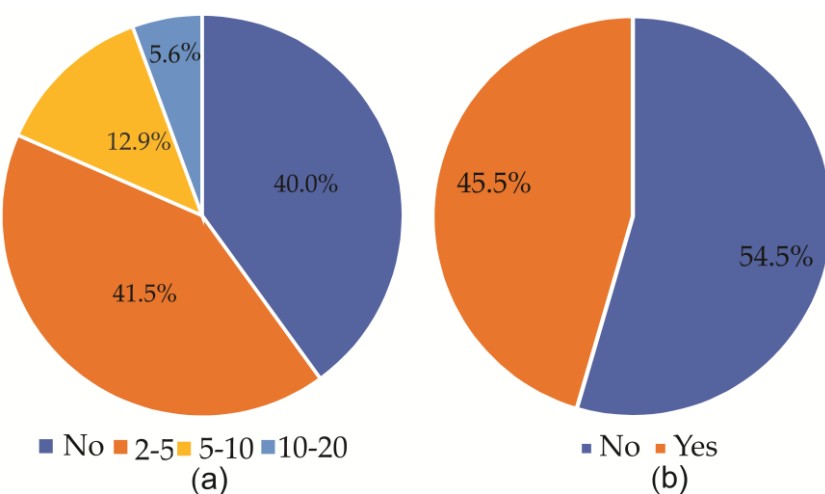

**Figure 3.** Purchases of construction and building materials and products in percentages: (**a**) before the pandemic (Q16) and (**b**) during the pandemic and world crisis (Q19).

Only 4% of the interviewees reported not purchasing any building supplies or products before 2020. Compared to the prior time, an even lower percentage of consumers refrained from purchasing these goods throughout the pandemic and the ongoing global crisis (Figure 4). The methodology of selecting the products of interest was altered during the observed periods, in that more respondents opted for a fair relationship between quality and price earlier, while a larger proportion of respondents chose to seek assistance from a contractor or knowledgeable acquaintance after the beginning of the crisis. The smallest percentages of consumers purchase inexpensive goods, expensive ones, or those made by well-known companies.

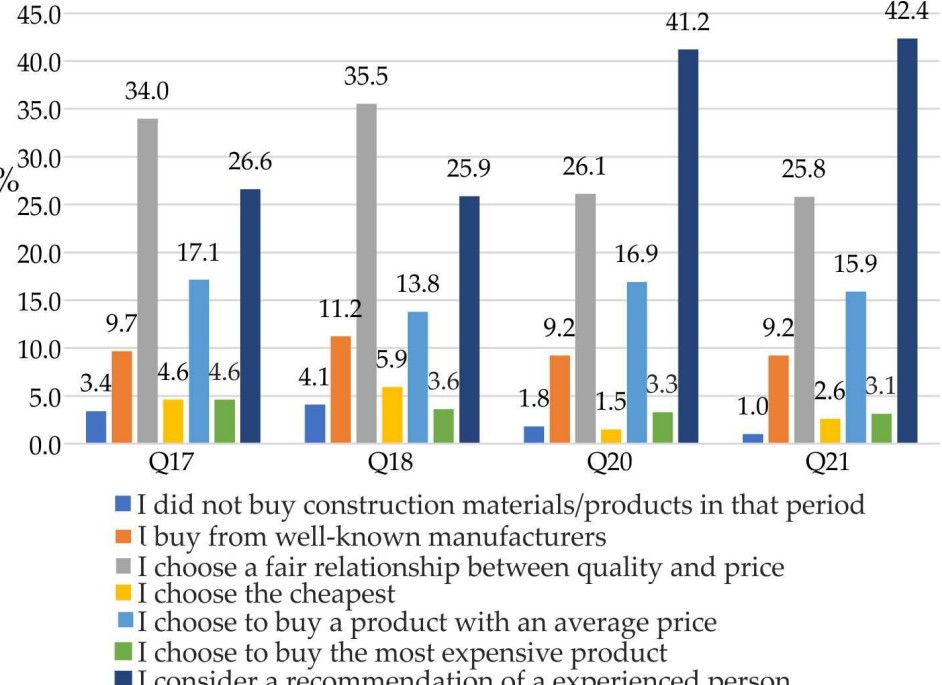

**Figure 4.** Based on what the respondents choose: construction materials before the pandemic and world crisis (Q17), construction products before the pandemic and world crisis (Q18), construction materials during the pandemic and world crisis (Q20), and construction products during the pandemic and world crisis (Q21).

### 3.3. An Attitude toward Novel and Ecological Materials

Likert's 7-level scale was used in this block of questions to reveal the attitude of the correspondents on the influences that drove the choice of using novel eco-labelled materials (Figure 5). Again, the 4 (the average) and 5 (one level above the average) marks were given to most of the questions. The only exception is that the knowledge of the benefits appeared more important than other parameters. If using environmentally friendly building materials and products does not result in a price increase or specific alterations to the apartment or home they live in, and they are aware of the advantages, the correspondents are largely driven to do so. Knowledge of the welfare of an eco-material/product appears critical in making this decision. As a result, the producers should be more forthcoming on this subject. Another problem is that there are not many environmentally friendly products available on the Serbian market, at least not those that are specifically marked as such.

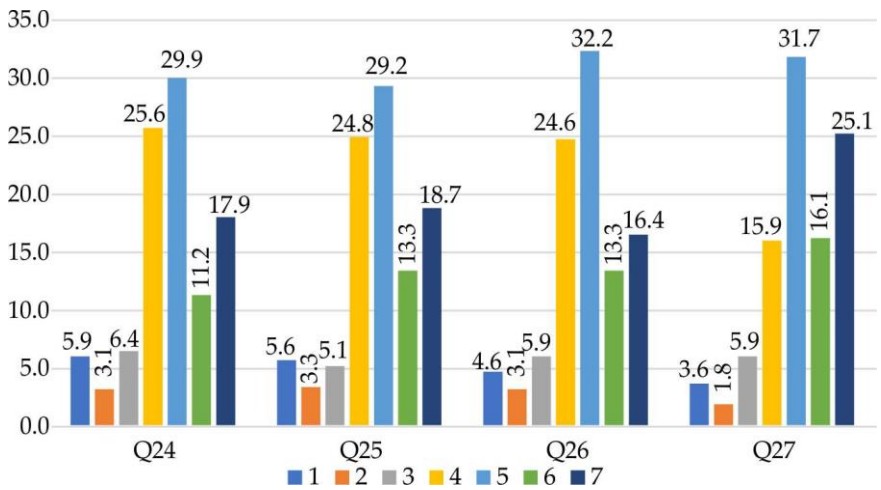

**Figure 5.** Novel materials usage willingness, as per the 7-levels Likert's scale (Willingness to use novel materials—Q24, the effect of price—Q25, The scope of the work required—Q26, and The knowledge of the benefits—Q27).

Approximately 6% of those polled stated that they are not interested in using any environmentally friendly materials (Q24). Given that the impact on the health of the majority of newly produced chemicals in the construction and building sector has yet to be assessed [42], it is not surprising that some people are cautious about using eco-labelled products. These conclusions are closely aligned with the UN Sustainable Development Goals for 2030 [43]. In line with other studies and surveys conducted in the industrial sector, it is also reasonable to assume that the increase in a product's price causes more information to be required by potential consumers [44] and that a price increase can act as a barrier to adopting eco-labelled products [45]. Understanding the key benefits of using environmentally friendly construction and building products by developing countries' construction firms and individual clients will increase the performance and returns on investments. Recognizing the potential benefits will improve the demand–supply dynamics in this market sector [29].

The final series of questions was used to gauge how willing the participants were to employ ecologically friendly materials. The respondents presumably do not currently use novel materials (78.5%), Figure 6. In addition, a large share (12.5%) of participants are not aware if they use one. The share of these answers is grouped into non-shaped and shaped construction materials or products. Participants from developing countries who use novel products (Figure 6) primarily purchased construction products (58.14%), while the remainder use non-shaped materials (41.86%).

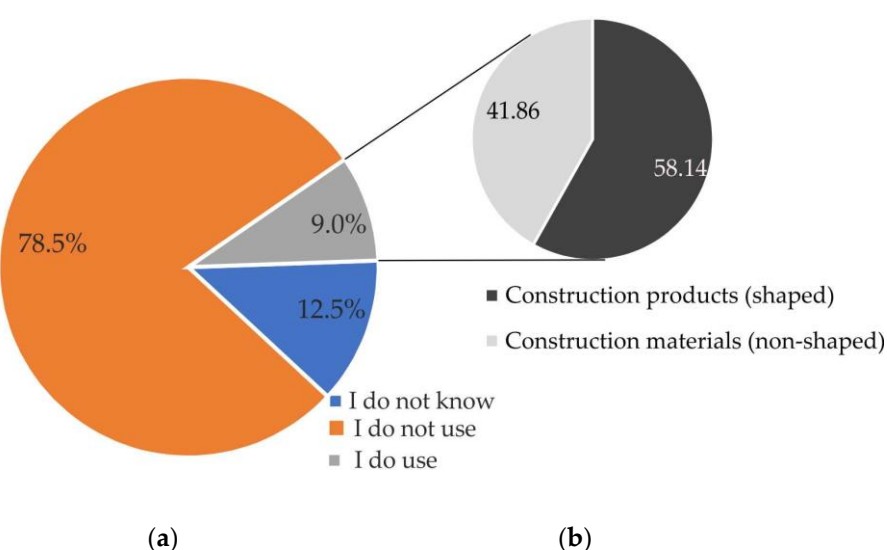

**Figure 6.** (**a**) Novel materials' current usage (Q28) and (**b**) which of the products are used (Q28).

*3.4. Influence of Sociodemographic Characteristics on Building Materials and Products Purchase (Statistical Analysis)*

The sample consisted of 391 respondents who answered all of the questions. This sample size permeates the smallest sample size necessary to describe the subject country and the rule of 10 (at least 10 cases for each item in the instrument being used) quite generously [46,47].

The majority of those who completed the questionnaire belonged to the middle-aged group, lived in Belgrade, and owned their place of residence and a rest house/another apartment in an older building. Most respondents were female and had a higher education level. Respondents with a higher degree of education were mostly those that were satisfied with their earnings. Furthermore, income satisfaction was statistically significantly correlated to income changes since the beginning of the COVID-19 pandemic and the global crisis among apartment dwellers. The rest of the home-owners were mostly retired people whose income did not change, who had not recently moved, and who lived in their own homes of some higher age. The majority of the respondents have not recently changed their residence because they live in their own homes.

Spearman's correlation analysis revealed several important relations between the independent, extended sociodemographic (Q1–Q14) and dependent variables (Q15–Q28). Gender, age, and whether the participant was renting or owning their home were the factors statistically significantly correlated to most of the other variables (Table 3). Women are less likely to live in larger cities or change their place of residence, and lived in older buildings; men tend to gain a higher level of education. When increasing age, the respondents rent less often, live in bigger cities and older buildings, and own an old rest house. Those who possess a place of residence tend to own a rest house. Income satisfaction is correlated to the level of education and profession.

Respondents who believed that a product's price indicates its quality stated that prices changed significantly during the pandemic (Table 4). The most prominent differences suggested that the socio-economic change due to the pandemic had a significant impact on the construction and building materials and products markets in developing countries, influencing the way these products are chosen. The respondents that believe the price is a good reflection of the product's quality are more interested in ecological products, but would like to obtain more information on the subject.

**Table 3.** Spearman's correlation coefficients of sociodemographic data and their statistical significance (Q1–Q14).

| | Age | Gender | Inhabit.[1] | Educat. | Profession | Income Satisf. | Income Change | Moving | Ap./House | Rented or not | Build. Age | Rest House or not | Rest House Age |
|---|---|---|---|---|---|---|---|---|---|---|---|---|---|
| Responding method | 0.04 | 0.21 ** | 0.01 | −0.02 | −0.03 | 0.08 | −0.23 | −0.09 | −0.07 | 0.08 | −0.07 | 0.02 | −0.02 |
| Age | | −0.03 | 0.12 * | −0.04 | 0.11 | 0.02 | −0.10 | 0.28 ** | −0.04 | 0.38 ** | 0.28 ** | 0.20 ** | 0.21 ** |
| Gender | | | −0.12 ** | 0.17 ** | 0.12 * | 0.05 | −0.02 | −0.12 * | 0.06 | −0.11 | −0.19 ** | 0.04 | 0.05 |
| Inhabit. | | | | −0.13 * | −0.11 | −0.01 | 0.05 | −0.04 | −0.10 | 0.08 | 0.10 | 0.10 | 0.06 |
| Educat. | | | | | 0.38 ** | 0.15 * | −0.08 | −0.01 | 0.01 | −0.07 | −0.04 | −0.02 | 0.02 |
| Profession | | | | | | 0.11 * | 0.01 | −0.01 | −0.09 | −0.09 | 0.04 | 0.03 | 0.03 |
| Income satisf. | | | | | | | 0.23 ** | −0.01 | −0.16 ** | 0.14 * | 0.01 | 0.05 | 0.05 |
| Income change | | | | | | | | 0.06 | 0.00 | 0.03 | 0.11 * | 0.05 | 0.07 |
| Moving | | | | | | | | | 0.21 ** | 0.34 ** | 0.27 ** | 0.06 | 0.10 |
| Ap./house | | | | | | | | | | 0.09 | −0.07 | 0.03 | 0.01 |
| Rented or not | | | | | | | | | | | 0.18 ** | 0.24 ** | 0.25 ** |
| Build. age | | | | | | | | | | | | −0.01 | 0.05 |
| Rest house or not | | | | | | | | | | | | | 0.85 ** |

[1] Abbreviations: Inhabit, number of inhabitants; Educat, education level; Income satisfy; income satisfaction, Ap./house, apartment or house, build. Age, Age of the building. *—Correlation is significant at the 0.05 level (2-tailed), **—Correlation is significant at the 0.01 level (2-tailed).

**Table 4.** Spearman's correlation coefficients of the responses relevant to the study with statistical significance (Q15–Q28).

| | CB[1] Purchase BP | CBM Choice BP | CBP Choice BP | CB Purchase DPC | CBM Choice DPC | CBP Choice DPC | Price/ Quality DPC | Prices Change of CBM (DPC) | Eco-Products acc. | Price Effect | Scope of Work | Knowledge | Eco-CB |
|---|---|---|---|---|---|---|---|---|---|---|---|---|---|
| Price/quality | −0.01 | −0.01 | −0.02 | 0.11 * | −0.03 | −0.02 | 0.26 ** | 0.14 ** | 0.24 ** | 0.07 | 0.10 | 0.12 * | 0.06 |
| CB purchase BP | | −0.47 ** | −0.42 ** | 0.30 ** | −0.28 ** | −0.01 | 0.16 ** | 0.02 | 0.02 | 0.05 | 0.00 | 0.00 | 0.02 |
| CBM choice BP | | | 0.83 ** | −0.26 ** | 0.48 ** | 0.49 ** | 0.10 | −0.16 ** | −0.09 | −0.06 | −0.12 * | −0.07 | 0.01 |
| CBP choice BP | | | | −0.31 ** | 0.57 ** | 0.58 ** | −0.19 ** | −0.21 ** | −0.08 | −0.05 | −0.11 * | −0.09 | −0.02 |
| CB purchase DPC | | | | | −0.67 ** | −0.64 ** | 0.04 | 0.18 ** | 0.13 * | 0.01 | 0.06 | 0.06 | 0.16 ** |
| CBM choice DP | | | | | | 0.92 ** | −0.14 * | −0.14 * | −0.12 * | −0.03 | −0.11 * | −0.10 | 0.01 |
| CBP choice DPC | | | | | | | −0.13 * | −0.15 ** | −0.12 * | −0.04 | −0.09 | −0.10 | 0.03 |
| Price/quality DPC | | | | | | | | 0.33 ** | 0.22 ** | 0.24 ** | 0.26 ** | 0.12 * | −0.11 |
| Prices change of CB | | | | | | | | | 0.17 ** | 0.30 ** | 0.20 ** | 0.27 ** | 0.04 |
| Eco-products acc. | | | | | | | | | | 0.34 ** | 0.36 ** | 0.36 ** | −0.01 |
| Price effect | | | | | | | | | | | 0.68 ** | 0.52 ** | −0.01 |
| Scope of work | | | | | | | | | | | | 0.55 ** | 0.01 |
| Knowledge | | | | | | | | | | | | | 0.14 * |

[1] Abbreviations: CB—construction and building materials and products: CBM, construction and building materials; CBP—construction and building products, Knowledge—Need for knowledge of the benefits of using eco-products, BP—before the pandemic, DPC—during the pandemic and crisis, acc.—acceptance, *—Correlation is significant at the 0.05 level (2-tailed), **—Correlation is significant at the 0.01 level (2-tailed).

Among the sociodemographic parameters observed (Table 5), the main influence on the market trends is seen in education level, place of residence (apartment or a house), gender, and whether a respondent owns a rest house. Better-educated people believe that the price speaks for the quality, and they are more interested in details related to ecological products before accepting their application. Well-educated answerers generally expressed a higher interest in ecological products. In addition, they claim that they rarely seek the advice of a more experienced person before making a purchase. The scope of work required to implement a novel material appeared to be something that more often considers men, those that are satisfied with their salaries, and those who own a house for a living. The acceptance of eco-products and recent changes in income did not show a significant correlation to any studied data. Gender reflected differences in the way of choosing the products of interest, but both reflected an equal interest in environmental issues, as shown in other studies [48,49]. Before the pandemic, higher education was associated with purchasing more expensive materials and products, but this has changed after the pandemic. The calculations revealed that income satisfaction, profession, living in an apartment or a house, owning a rest house, and changing place of residence were all highly correlated with general price–quality perceptions. The situation has changed with the pandemic and the ongoing global crises because only the mature population living in older buildings thought that price no longer correlates to quality. The elderly respondents who had not recently moved, who lived in their residences, and owned a rest house were the ones who spent the most money on construction and building materials before the crisis. In the more recent period, buyers were more often men owning a house and a cottage in smaller cities. Significant price changes since the beginning of March 2020 are seen by all groups of respondents. Those who own a rest house purchase materials and products regularly, and the older the building they saw a decrease in the price/quality ratio during the crisis.

### 3.5. Factors Influencing Decisions on the Use of Novel Materials

Respondents who have experienced price increases after 2020 are more interested in accepting eco-labelled products and emphasize the importance of factors such as the scope of work required, the required prior knowledge, and the impact of cost, by declining importance respectively (Table 4). The conclusion is similar to that in Egypt about accepting novel sustainable value management implementation [31]. Factors associated to ecological materials and products were shown to be unrelated to any of the following sociodemographic parameters listed in Table 5. The well-educated respondents were mostly those who purchased eco-labelled products (Table 5). The price of these products, the scope of the work required, and knowledge of the benefits, according to them, are the key factors influencing their acceptance. Those who are content with their earnings emphasized that knowledge was the most important factor in their decision-making process. In contrast to those who live in apartments, respondents who live in houses believe that the scope of the required work is not significant.

In this sector, both important steps (awareness and adoption) need to be addressed more clearly by the manufacturers and suppliers in Serbia. Only awareness is not considered sufficient to act [50]. The existence of attitude–behavior gaps was also demonstrated by other investigations in the building sector [51]. A more specialized awareness campaign may be able to achieve the considerable benefits of changing behavior. Researchers who study building materials and company management both have significant knowledge gaps when it comes to sustainability, which might be a major hurdle. One of the fundamental tenets is that the value chain of building materials, from cradle to grave, defines their sustainability. The research and practice have not yet extensively embraced this way of evaluating sustainability since financial performance is often considered the primary factor, while climate change is neglected [52].

**Table 5.** Correlation matrices of sociodemographic data and responses relevant to the study with statistical significance.

| | Price/ Quality | CB [1] Purchase BP | CBM Choice BP | CBP Choice BP | CB Purchase DPC | CBM Choice DPC | CBP Choice DPC | Price/ Quality DPC | Prices Change of CBM (DPC) | Eco-Products acc. | Price Effect | Scope of Work | Knowledge | Eco-CB |
|---|---|---|---|---|---|---|---|---|---|---|---|---|---|---|
| **Responding method** | 0.10 | −0.08 | 0.08 | 0.06 | 0.05 | 0.08 | 0.11 * | −0.02 | 0.10 | 0.02 | 0.02 | −0.01 | 0.10 | 0.08 |
| **Age** | 0.00 | 0.19 ** | −0.11 * | −0.10 | −0.01 | 0.03 | 0.02 | −0.14 * | 0.05 | −0.08 | 0.03 | 0.02 | 0.06 | 0.10 |
| **Gender** | 0.10 | 0.10 | −0.16 ** | −0.19 ** | 0.12 * | −0.16 * | −0.16 ** | 0.10 | 0.04 | 0.08 | 0.07 | 0.11 * | 0.03 | 0.07 |
| **Inhabit.** | −0.04 | 0.04 | −0.01 | −0.01 | −0.12 * | 0.10 | 0.07 | −0.08 | 0.02 | 0.12 | −0.03 | −0.11 * | 0.04 | −0.03 |
| **Educat.** | 0.12 * | 0.06 | −0.14 ** | −0.12 * | −0.04 | 0.03 | 0.01 | 0.04 | 0.05 | 0.07 | 0.18 ** | 0.14 * | 0.20 ** | 0.17 ** |
| **Profession** | 0.18 ** | 0.03 | −0.01 | −0.01 | −0.01 | 0.08 | 0.07 | −0.03 | 0.02 | 0.08 | 0.06 | 0.02 | 0.11 * | 0.09 |
| **Income satisf.** | 0.39 ** | 0.07 | 0.04 | −0.01 | 0.06 | 0.02 | 0.03 | 0.07 | −0.01 | 0.10 | 0.02 | 0.04 | 0.12 * | 0.10 |
| **Income change** | 0.00 | −0.08 | −0.01 | −0.01 | 0.06 | −0.05 | −0.07 | 0.02 | 0.00 | 0.04 | −0.06 | 0.01 | 0.00 | −0.03 |
| **Moving** | −0.16 ** | −0.15 ** | −0.05 | −0.03 | 0.09 | −0.09 | −0.06 | 0.00 | 0.02 | −0.06 | 0.04 | 0.01 | 0.07 | 0.10 |
| **Ap./house** | −0.16 ** | 0.09 | −0.17 ** | −0.17 ** | 0.12 * | −0.22 ** | −0.24 ** | 0.04 | 0.01 | −0.05 | −0.04 | −0.11 * | −0.02 | 0.02 |
| **Rented or not** | 0.05 | 0.21 ** | −0.08 | −0.07 | 0.02 | −0.05 | −0.06 | −0.04 | 0.10 | −0.01 | −0.06 | −0.04 | 0.02 | 0.08 |
| **Build. age** | −0.08 | 0.01 | 0.02 | 0.05 | −0.04 | 0.05 | 0.04 | −0.17 ** | −0.01 | −0.03 | −0.02 | −0.04 | 0.03 | 0.08 |
| **Rest house or not** | 0.14 * | 0.16 ** | −0.11 * | −0.12 * | 0.15 ** | −0.15 ** | −0.10 | 0.02 | 0.04 | −0.05 | 0.01 | 0.03 | −0.04 | 0.10 |
| **Rest house age** | 0.15 ** | 0.14 * | −0.11 * | −0.13 ** | 0.08 | −0.13 * | −0.09 | 0.02 | 0.03 | −0.04 | 0.05 | 0.08 | 0.00 | 0.10 |

[1] Abbreviations: Inhabit.—Number of inhabitants, Educat.—Education level, Income satisf.—Income satisfaction, Ap./house—Apartment or house, Build. age—Age of the building, CB—construction and building materials and products, CBM—construction and building materials, CBP—construction and building products, Knowledge—Need for knowledge of the benefits of using eco-products, BP—before the pandemic, DPC—during the pandemic and crisis, acc.—acceptance, *—correlation is significant at the 0.05 level (2-tailed), **—correlation is significant at the 0.01 level (2-tailed).

## 4. Conclusions

The main goal of this study was to look into the impact of the crisis beginning in 2020 on socio-economic issues in Serbia as an example of a developing country, which was related to building materials and product usage and purchase. The study fills a literature gap and aids in understanding the impact of the crisis on salaries, market prices, and material consumption in the construction sector. It also provides a general review of environmental awareness and the acceptability of emerging sustainable products. The main conclusions of the study are as follows:

1. The respondents' common experience was that the price of a product is averagely or slightly dependent on its quality, that the price/quality ratio has changed during the crisis, and that the prices in the construction and building industry have increased. Before the start of the crisis, the respondent's decision-making regarding purchases was largely influenced by the product's price. This has changed since 2020 because professional advice has been sought in this regard;

2. The most prominent statistically significant correlations suggested that the pandemic had a significant impact on the construction and building materials and products markets in Serbia, influencing the product selection criteria. The older age groups were the purchasers of reasonably priced products before the crisis. Respondents with higher education were a majority of those who purchased eco-labelled products. Higher education was associated with purchasing more expensive materials and products before the pandemic, but not after the outbreak. Respondents who had not recently moved and those who live in houses spent the most money on buying these products and showed less trust in a reasonable price/quality relationship. The most frequent purchasers of the materials and products in this sector were apartment and house owners;

3. To effectively handle climate change, the local government must be aware of citizen preferences. Awareness and action analyses are crucial for understanding the social and behavioral context of the issue and creating workable local solutions.

4. Professionals need to adopt a more environmentally conscious mindset. To carry out the technological and strategic changes in the construction industry, upgrading new skills and competencies is required. A further improvement in the sharing of relevant information with potential customers is a requirement.

The limitations of the study are in a relatively low number of respondents. This is a preliminary study, and consumer behavior in this area must be studied to understand the factors that influence this behavior. In a future study, the Theory of Planned Behaviour (TPB) will be used for the prediction of the construction materials purchases of Serbian consumers. Cross-sectional data will be performed using collected data through a self-administered survey. The appropriateness of theory and conceptual framework will be tested using structural equation modeling (SEM).

**Author Contributions:** Conceptualization, M.V.V., G.G., M.T., M.P. and L.P.; Survey questions formulation, M.V.V., G.G., M.D., S.Ž., M.T., M.P. and L.P.; methodology, M.V.V., G.G., M.P. and L.P.; software, M.V.V.; M.P. and L.P.; writing—original draft preparation, M.V.V. and L.P; writing—review and editing, M.V.V., G.G., M.D., S.Ž., M.T. and M.P., visualization, M.V.V.; supervision, M.V.V. and L.P.; Economic issues discussion, S.Ž. All authors have read and agreed to the published version of the manuscript.

**Funding:** This research was funded by the Ministry of Science, Technological Development and Innovation of the Republic of Serbia, Contracts No. 451-03-68/2022-14/200012, 451-03-47/2023-01/200017, and 451-03-47/2023-01/200051.

**Institutional Review Board Statement:** Not applicable.

**Informed Consent Statement:** Informed consent was obtained from all subjects involved in the study.

**Data Availability Statement:** The data is contained within the article. Additional data are available upon request from the corresponding author.

**Conflicts of Interest:** The authors declare no conflict of interest.

**Appendix A**

A study on the use of building materials in developing countries before and after the pandemic: A socio-economic analysis (List of the questions)

Q1. The method of responding to the survey:

- Smartphone
- Desktop
- Tablet
- In paper

Q2. What is your age group?

- 18–30
- 30–40
- 40–50
- 50–60
- 60–70
- 70–80

Q3. What gender are you?

- Male
- Female
- None of the above

Q4. How many inhabitants are there in the place where you live?

- Under 100,000
- Between 100,000 and 300,000
- Between 300,000–500,000
- Between 500,000 and 800,000
- Between 800,000–1,000,000
- Between 1,000,000 and 2,000,000
- Over 2,000,000

Q5. What is your final education level?

- Primary school
- High School
- College
- Researcher/Doctor of Science
- Professor

Q6. What is your profession/job description?

- Unemployed
- Manual worker
- Office work
- Laboratory technician
- Medical worker
- Craftsman
- Student
- Artist
- Engineer
- Manager
- Scientist
- Professor
- Retired
- Other

Q7. How satisfied are you with your salary concerning the work you do? (the optional question)
From very dissatisfied to very satisfied
(Scale 1–7)
Q8. Has your income changed since the crisis (pandemic) began?

- Incomes have decreased
- They haven't changed
- Incomes have increased
- Not applicable (retired, non-employed)

Q9. When was the last time you changed your place of residence?

- In the last 5 years
- During the last 10 years
- 20 years ago
- 30 years ago
- More than 40 years ago
- Never

Q10. Do you live in an apartment or a house?

- Apartment
- A house

Q11. Are you renting the space you live in or is it owned by you or your family?

- I'm renting
- I live in mine/our apartment/house

Q12. When was the building/house (you currently live in) built?

- In the last 5 years
- During the last 10 years
- 20 years ago
- 30 years ago
- 40 or more years ago

Q13. Do you own a cottage, a rest private house, or more than one apartment?

- Yes
- No

Q14. If you own a cottage, a rest private house, or more than one apartment, when was it built?

- In the last 5 years
- During the last 10 years
- 20 years ago
- 30 years ago
- 40 years ago, or more
- Not applicable

Q15. To what extent do you believe that the price of a product speaks of its quality?
(Likert's scale 1–7)
1—very low, 7—extremely high
Q16. In the period before the pandemic, did you buy construction materials or products (glue, varnish, paint, wall paint, cement, ceramic tiles, sanitary equipment, bricks, tiles, and floor coverings)?

- Yes, about 2–5 years ago.
- Yes, about 5–10 years ago.
- Yes, over about 10–20 years.
- No

Q17. Based on what did you choose construction materials in the period before the pandemic (glue, varnish, paint, wall paint, cement ...)?

- You choose to buy the most expensive product
- You choose to buy a product whose price is average
- You choose a fair relationship between quality and price
- You choose the cheapest
- You listen to the recommendation of a contractor or a friend/acquaintance you trust
- You buy from familiar manufacturers
- I did not buy construction materials during that period

Q18. Based on what did you choose construction products in the period before the pandemic (ceramic tiles, sanitary equipment, bricks, tiles, floor coverings ...)?

- You choose to buy the most expensive product
- You choose to buy a product whose price is average
- You choose a fair relationship between quality and price
- You choose the cheapest
- You listen to the recommendation of a contractor or a friend/acquaintance you trust
- You buy from familiar manufacturers
- I did not buy construction materials during that period

Q19. In the period after the beginning of the pandemic (March 2020), did you buy glue, varnish, paint, wall paint, cement, ceramic tiles, sanitary equipment, bricks, tiles, and floor coverings?

- Yes
- No

Q20. Based on what did you choose construction materials (glue, varnish, paint, wall paint, cement ...) in the period after the beginning of the pandemic (March 2020)?

- You choose to buy the most expensive product
- You choose to buy a product whose price is average
- You choose a fair relationship between quality and price
- You choose the cheapest
- You pay attention to the recommendation of a contractor or a friend/acquaintance you trust
- You buy from familiar manufacturers
- I did not buy construction materials during that period

Q21. Based on what did you choose construction products in the period during the 2020 crisis (ceramic tiles, sanitary equipment, bricks, tiles, floor coverings ...)?

- You choose to buy the most expensive product
- You choose to buy a product whose price is average
- You choose a fair relationship between quality and price
- You choose the cheapest
- You listen to the recommendation of a contractor or a friend/acquaintance you trust
- You buy from familiar manufacturers
- I did not buy construction materials during that period

Q22. To what extent has the way you choose products according to the price/quality ratio changed since the 2020 crisis?
(Likert's scale of 1–7)
1—very low, 7—extremely high

Q23. If you bought construction materials and/or products in the period before and after the 2020 crisis, to what extent do you have the impression that prices have changed?
(Likert's scale of 1–7)
1—very low, 7—extremely high

Q24. To what extent are you willing to accept a newer type of product compared to those traditionally used (nanocoating, cement-based geopolymers, fly ash-based cement, concrete based on geopolymers, self-healing concrete, concrete block instead of brick, concrete reinforced with bamboo, lightweight block of large dimensions, ceramic tiles of large dimensions, etc.?)

(Likert's scale of 1–7)

1—very low, 7—extremely high

Q25. To what extent would the price affect the acceptability of switching to some type of better environmental material/product in your household?

(Likert's scale of 1–7)

1—very low, 7—extremely high

Q26. To what extent would the scope of work be required to affect the acceptability of switching to some type of better environmental material/product in your household?

(Likert's scale of 1–7)

1—very low, 7—extremely high

Q27. To what extent would adequate knowledge of the benefits of new environmental materials/products affect the transition to that material/product in your household?

(Likert's scale of 1–7)

1—very low, 7—extremely high

Q28. Do you use any of the innovative materials/products from this sector and which ones?

- Indicate: ___________
- I do not use

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
