# Peer review of "Socio-Economic Analysis of the Construction and Building Materials’ Usage—Ecological Awareness in the Case of Serbia"

_sustainability, doi:10.3390/su15054080_

Round 1
Reviewer 1 Report
Thanks for the opportunity to review. I think that the data and the findings of this study are original and very interesting; however, I believe that the authors need to work more on the framing of this paper.
1. The potential impact of the COVID pandemic needs more explanations in the beginning. Initially, I was confused about how the spread of disease would affect people's choice of construction materials. I later understood that it is the socioeconomic change due to the pandemic that affected people's choice of the material. If that is true, it is socioeconomic change in Serbia, not the COVID pandemic itself, that is important in this study. This needs to be clear from the beginning.
2. For the readers who are not familiar with Serbia, please explain briefly the social context in Serbia and what was changed during the time of the pandemic. Also, readers need to know how much environment-friendly products are accepted by ordinary Serbians and how much they are more expensive than traditional products in Serbia.
3. The title can be reconsidered. Serbia is not representing the whole developing countries and there is only one case in this study; therefore, I believe that the authors do not have to mention developing countries in the title.
4. Overall, I feel that the paper is talking about many factors here and there and is not clear what the key contribution is. For example, the line 113 discuses the aims of the study, and the line 389 lays out totally different goals of this study. I feel that the focus of this paper is keep changing, which makes the paper hard to understand.
I hope to see a revised version of this interesting study. Thanks!
Author Response
Thank you for your time and effort in evaluating our manuscript. The responses to every discussion point are written in plain text after your comments which are in italic font. We hope that we fulfilled the task to frame the paper in a more comprehensive way.

Reviewer 2 Report
1. The abstract has to include the summary of findings also along with the research methodology, objective, research gap, and conclusion.
2. Table 1, Age Group (Q2), Frequency = 390, the summation should be 391.
3. The sampling number of N=391 out of the population of? what is the confidence level?
Author Response

(The authors gave the same response as above.)

Reviewer 3 Report
Although this manuscript presents an interesting topic, this submission must conform to the requirements of a scientific journal. If the following points are resolved and revisions are made, it can be considered for publication.
Abstract: First, start by explaining the significance of your topic by providing a line or two about the importance of conducting a socioeconomic analysis of construction and building materials usage.
Lines 26-29: In the abstract, it is not necessary to state the number of respondents or the age range of respondents, gender etc. It is not appropriate for detailed demographic information to be included in the abstract section.
In general, the significance of the main topic and the knowledge gap in the literature should be briefly explained, as should the methods employed in the study, the findings, and the implications of the findings for future research. These are the fundamental guidelines that the Abstract should adhere to. It is strongly suggested that the Abstract section be revised taking these guidelines into consideration.
Introduction: The Introduction section is far lengthy, and it includes sections that ought to be included in a well-organized and separate literature review section. Move some of the sections that incorporate previous research that is related into a literature review section. The literature review should highlight the knowledge gap and explain how the current study addresses the knowledge gap. The knowledge gap should also be brought briefly to the reader's attention in the introduction section.
In addition, there are other aspects of the Introduction that need to be addressed, including the following:
Lines 47-51: "One of the many...." These statements ought to be supported by a reference, or references.
Lines 54-55: Instead of "opinions and feelings of people", you should use a language such as "perspectives and perceptions of respondents" since it has a more academic sound.
Lines 71-73: "The reason behind....": This statement ought to be supported by a reference, or references.
Data Collection and Methodology: A literature review is the first step in the methodology of a questionnaire study.
Provide a research flowchart (maybe a Figure) that outlines the methodology used to conduct the study. Then offer a step-by-step explanation of the research flow chart, which should include elements like a literature review, the design of a questionnaire, data collection, and data analysis....
Regarding the format and methodology, kindly consult some more current and insightful questionnaire research, such as:
https://doi.org/10.3390/buildings11120618
Furthermore, please address the following points in the research methodology section:
How were the questions in the survey and the questionnaire structure validated?
Why was the random sampling technique used for sampling?
How do you justify your sample size of 391?
Why was a 7-point Likert scale used instead of for example 5?
The section on research methodology should also include a description of the statistical tests used for data analysis. This is missing.
Again, kindly consult some more current and insightful questionnaire research, such as:
https://doi.org/10.3390/buildings11120618
Conclusions: Limitations of this study?
Author Response

(The authors gave the same response as above.)

Round 2
Reviewer 1 Report
I think my concerns about the previous manuscript have been addressed in the current version. I do not have any concerns. Thanks for the opportunity to review.
Author Response
Thank you for the positive opinion about our work.
Reviewer 3 Report
I would like to thank the Authors for their efforts in revising the manuscript; nonetheless, there are still a few concerns, which are as follows:
1) Calculating the size of the sample can be done using a scientific sample size formula that is available in the applicable research literature and should be indicated in the manuscript. In addition, the population size, sample size, margin of error realized etc. in the sample size formula should all be stated in the appropriate section of the manuscript in which the sample size of 391 is justified.
2) The section on statistical analysis appears to be a little bit too brief. Please provide some further information on the tests that were carried out. It could be useful to include a few formulas with references.
3) Despite the fact that the seven-point Likert scale has been referenced, it should still be clarified why the seven-point scale was chosen rather than the five-point Likert scale and stated in the "text" of the manuscript.
4) In the conclusion section of the manuscript, the limitations of this study that were mentioned in the review report should be explicitly stated.
Author Response
Dear reviewer, thank you for your time and effort to evaluate our work again. The responses are attached.
